# METROPOLIS-HASTINGS VIEW ON VARIATIONAL INFERENCE AND ADVERSARIAL TRAINING

## ABSTRACT

In this paper we propose to view the acceptance rate of the Metropolis-Hastings algorithm as a universal objective for learning to sample from target distribution – given either as a set of samples or in the form of unnormalized density. This point of view unifies the goals of such approaches as Markov Chain Monte Carlo (MCMC), Generative Adversarial Networks (GANs), variational inference. To reveal the connection we derive the lower bound on the acceptance rate and treat it as the objective for learning explicit and implicit samplers. The form of the lower bound allows for doubly stochastic gradient optimization in case the target distribution factorizes (i.e. over data points). We empirically validate our approach on Bayesian inference for neural networks and generative models for images.

## 1 INTRODUCTION

Bayesian framework and deep learning have become more and more interrelated during recent years. Recently Bayesian deep neural networks were used for estimating uncertainty (Gal & Ghahramani, 2016), ensembling (Gal & Ghahramani, 2016) and model compression (Molchanov et al., 2017). On the other hand, deep neural networks may be used to improve approximate inference in Bayesian models (Kingma & Welling, 2014).

Learning modern Bayesian neural networks requires inference in the spaces with dimension up to several million by conditioning the weights of DNN on hundreds of thousands of objects. For such applications, one has to perform the approximate inference – predominantly by either sampling from the posterior with Markov Chain Monte Carlo (MCMC) methods or approximating the posterior with variational inference (VI) methods.

**MCMC** methods provide the unbiased (in the limit) estimate but require careful hyperparameter tuning especially for big datasets and high dimensional problems. The large dataset problem has been addressed for different MCMC algorithms: stochastic gradient Langevin dynamics (Welling & Teh, 2011), stochastic gradient Hamiltonian Monte Carlo (Chen et al., 2014), minibatch Metropolis-Hastings algorithms (Korattikara et al., 2014; Chen et al., 2016). One way to address the problem of high dimension is the design of a proposal distribution. For example, for the Metropolis-Hastings (MH) algorithm there exists a theoretical guideline for scaling the variance of a Gaussian proposal (Roberts et al., 1997; 2001). More complex proposal designs include adaptive updates of the proposal distribution during iterations of the MH algorithm (Holden et al., 2009; Giordani & Kohn, 2010). Another way to adapt the MH algorithm for high dimensions is combination of adaptive direction sampling and the multiple-try Metropolis algorithm as proposed in (Liu et al., 2000). Thorough overview of different extensions of the MH algorithm is presented in (Martino, 2018).

**Variational inference** is extremely scalable but provides a biased estimate of the target distribution. Using the doubly stochastic procedure (Titsias & Lázaro-Gredilla, 2014; Hoffman et al., 2013) VI can be applied to extremely large datasets and high dimensional spaces, such as a space of neural network weights (Kingma et al., 2015; Gal & Ghahramani, 2015; 2016). The bias introduced by variational approximation can be mitigated by using flexible approximations (Rezende & Mohamed, 2015) and resampling (Grover et al., 2018).

**Generative Adversarial Networks** (Goodfellow et al., 2014) (GANs) is a different approach to learn samplers. Under the framework of adversarial training different optimization problems could be solved efficiently (Arjovsky et al., 2017; Nowozin et al., 2016). The shared goal of "learning to

sample" inspired the connection of GANs with VI (Mescheder et al., 2017) and MCMC (Song et al., 2017).

In this paper, we propose a novel perspective on learning to sample from a target distribution by optimizing parameters of either explicit or implicit probabilistic model. Our objective is inspired by the view on the acceptance rate of the Metropolis-Hastings algorithm as a quality measure of the sampler. We derive a lower bound on the acceptance rate and maximize it with respect to parameters of the sampler, treating the sampler as a proposal distribution in the Metropolis-Hastings scheme.

We consider two possible forms of the target distribution: unnormalized density (*density-based* setting) and a set of samples (*sample-based* setting). Each of these settings reveals a unifying property of the proposed perspective and the derived lower bound. In the density-based setting, the lower bound is the sum of forward and reverse KL-divergences between the true posterior and its approximation, connecting our approach to VI. In the sample-based setting, the lower bound admit a form of an adversarial game between the sampler and a discriminator, connecting our approach to GANs.

The closest work to ours is of Song et al. (2017). In contrast to their paper our approach (1) is free from hyperparameters; (2) is able to optimize the acceptance rate directly; (3) avoids minimax problem in the density based setting.

Our main contributions are as follows:

1. We introduce a novel perspective on learning to sample from the target distribution by treating the acceptance rate in the Metropolis-Hastings algorithm as a measure of sampler quality.
2. We derive the lower bound on the acceptance rate allowing for doubly stochastic optimization of the proposal distribution in case when the target distribution factorizes (i.e. over data points).
3. For sample-based and density-based forms of target distribution we show the connection of the proposed algorithm to variational inference and GANs.

The rest of the paper is organized as follows. In Section 2 we introduce the lower bound on the AR. Special forms of target distribution are addressed in Section 3. We validate our approach on the problems of approximate Bayesian inference in the space of high dimensional neural network weights and generative modeling in the space of images in Section 4. We discuss results and directions of the future work in Section 5.

## 2 ACCEPTANCE RATE FOR THE METROPOLIS-HASTINGS ALGORITHM

### 2.1 PRELIMINARIES

In MH algorithm we need to sample from target distribution $p(x)$ while we are only able to sample from proposal distribution $q(x' \,|\, x)$. One step of the MH algorithm can be described as follows.

1. sample proposal point $x' \sim q(x' \,|\, x)$, given previously accepted point $x$
2. accept $\begin{cases} x', & \text{if } \frac{p(x')q(x \,|\, x')}{p(x)q(x' \,|\, x)} > u, \quad u \sim \text{Uniform}[0, 1] \\ x, & \text{otherwise} \end{cases}$

If the proposal distribution $q(x' \,|\, x)$ does not depend on $x$, i.e. $q(x' \,|\, x) = q(x')$, the algorithm is called *independent* MH algorithm.

The quality of the proposal distribution is measured by acceptance rate and mixing time. Mixing time defines the speed of convergence of the Markov chain to the stationary distribution. The acceptance rate of the MH algorithm is defined as

$$\text{AR} = \mathbb{E}_\xi \min\{1, \xi\} = \int dx dx' p(x) q(x' \,|\, x) \min\left\{1, \frac{p(x')q(x \,|\, x')}{p(x)q(x' \,|\, x)}\right\}, \tag{1}$$

where

$$\xi = \frac{p(x')q(x \,|\, x')}{p(x)q(x' \,|\, x)}, \quad x \sim p(x), \quad x' \sim q(x' \,|\, x). \tag{2}$$

In case of independent proposal distribution we show that the acceptance rate defines a semimetric in distribution space between $p$ and $q$ (see Appendix A.2).

## 2.2 Optimizing the lower bound on acceptance rate

Although, we can maximize the acceptance rate of the MH algorithm (Eq. 1) directly w.r.t. parameters $\phi$ of the proposal distribution $q_\phi(x' \,|\, x)$, we propose to maximize the lower bound on the acceptance rate. As our experiments show (see Section 4) the optimization of the lower bound compares favorably to the direct optimization of the acceptance rate. To introduce this lower bound we first express the acceptance rate in terms of total variation distance.

**Theorem 1** *For random variable* $\xi = \frac{p(x')q(x \,|\, x')}{p(x)q(x' \,|\, x)}, x \sim p(x), x' \sim q(x' \,|\, x)$

$$\mathbb{E}_\xi \min\{1, \xi\} = 1 - \frac{1}{2}\mathbb{E}_\xi |\xi - 1| = 1 - \mathrm{TV}\left( p(x')q(x \,|\, x') \Big\| p(x)q(x' \,|\, x) \right), \qquad (3)$$

*where* $\mathrm{TV}$ *is the total variation distance.*

The proof of Theorem 1 can be found in Appendix A.1. This reinterpretation in terms of total variation allows us to lower bound the acceptance rate through the Pinsker's inequality

$$\mathrm{AR} \geq 1 - \sqrt{\frac{1}{2} \cdot \mathrm{KL}\left( p(x')q(x \,|\, x') \Big\| p(x)q(x' \,|\, x) \right)}. \qquad (4)$$

The maximization of this lower bound can be equivalently formulated as the following optimization problem

$$\min_\phi \mathrm{KL}\left( p(x')q_\phi(x \,|\, x') \Big\| p(x)q_\phi(x' \,|\, x) \right). \qquad (5)$$

In the following sections, we show the benefits of this optimization problem in two different settings — when the target distribution is given in a form of unnormalized density and as a set of samples. In Appendix C.5 and C.1 we provide the empirical evidence that maximization of the proposed lower bound results in the maximization of the acceptance rate.

## 3 Optimization of Proposal Distribution

From now on we consider only optimization problem Eq. 5 but the proposed algorithms can be also used for the direct optimization of the acceptance rate (Eq. 1).

To estimate the loss function (Eq. 5) we need to evaluate the density ratio. In the density-based setting unnormalized density of the target distribution is given, so we suggest to use explicit proposal distribution to compute the density ratio explicitly. In the sample-based setting, however, we cannot compute the density ratio, so we propose to approximate it via adversarial training (Goodfellow et al., 2014). The brief summary of constraints for both settings is shown in Table 1.

Table 1: Constraints for two settings of learning sampling algorithms

| Setting | Target distribution | Proposal distribution | Density Ratio |
|---|---|---|---|
| Density-based | given $\hat{p}(x) \propto p(x)$ | explicit model $q(x')$ | explicit |
| Sample-based | set of samples $X \sim p(x)$ | implicit model $q(x')$ 
 implicit model $q(x'|x)$ | learned discriminator |

The following subsections describe the algorithms in detail.

### 3.1 Density-based Setting

In the density-based setting, we assume the proposal to be an explicit probabilistic model, i.e. the model that we can sample from and evaluate its density at any point up to the normalization constant. We also assume that the proposal is reparameterisable (Kingma & Welling, 2014; Rezende et al., 2014; Gal, 2016).

During the optimization of the acceptance rate we might face a situation when proposal collapses to the delta-function. This problem usually arises when we use Markov chain proposal, for example, $q_\phi(x' \,|\, x) = \mathcal{N}(x' \,|\, x, \sigma)$. For such proposal we can obtain arbitrary high acceptance rate, making the $\sigma$ small enough. However, this is not the case for the independent proposal distribution $q_\phi(x' \,|\, x) = q_\phi(x')$. In Appendix B.1 we provide more details and intuition on this property of acceptance rate maximization. We also provide empirical evidence in Section 4 that collapsing to the delta-function does not happen for the independent proposal.

In this paper, we consider two types of explicit proposals: simple parametric family (Section 4.2) and normalizing flows (Rezende & Mohamed, 2015; Dinh et al., 2016) (Section 4.1). Rich family of normalizing flows allows to learn expressive proposal and evaluate its density in any point of target distribution space. Moreover, an invertible model (such as normalizing flow) is a natural choice for the independent proposal due to its ergodicity. Indeed, choosing the arbitrary point in the target distribution space, we can obtain the corresponding point in the latent space using the inverse function. Since every point in the latent space has positive density, then every point in the target space also has positive density.

Considering $q_\phi(x')$ as the proposal, the objective of optimization problem 5 takes the form

$$\mathcal{L}(p, q_\phi) = \mathrm{KL}\left( p(x')q_\phi(x) \middle\| p(x)q_\phi(x') \right) = \mathop{\mathbb{E}}_{\substack{x \sim p(x) \\ x' \sim q_\phi(x')}} \log \frac{p(x)q_\phi(x')}{p(x')q_\phi(x)}. \tag{6}$$

Explicit form of the proposal $q_\phi(x')$ and the target $p(x)$ distributions allows us to obtain density ratios $q_\phi(x)/q_\phi(x')$ and $p(x')/p(x)$ for any points $x, x'$. But to estimate the loss in Eq. 6 we also need to obtain samples from the target distribution $x \sim p(x)$ during training. For this purpose, we use the current proposal $q_\phi$ and run the independent MH algorithm. After obtaining samples from the target distribution it is possible to perform optimization step by taking stochastic gradients w.r.t. $\phi$. Pseudo-code for the obtained procedure is shown in Algorithm 1.

---

**Algorithm 1** Optimization of proposal distribution in density-based case

---

**Require:** explicit probabilistic model $q_\phi(x')$
**Require:** density of target distribution $\hat{p}(x) \propto p(x)$
   **while** $\phi$ not converged **do**
      sample $\{x'_k\}_{k=1}^K \sim q_\phi(x')$
      sample $\{x_k\}_{k=1}^K \sim p(x)$ using independent MH with current proposal $q_\phi$
      $\mathcal{L}(p, q_\phi) \simeq \frac{1}{K}\sum_{k=1}^K \log \frac{p(x_k)q_\phi(x'_k)}{p(x'_k)q_\phi(x_k)}$       ▷ approximate loss with finite number of samples
      $\phi \leftarrow \phi - \alpha\nabla_\phi \mathcal{L}(p, q_\phi)$       ▷ perform gradient descent step
   **end while**
   **return** optimal parameters $\phi$

---

Algorithm 1 could also be employed for the direct optimization of the acceptance rate (Eq. 1). Now we apply this algorithm for Bayesian inference problem and show that during optimization of the lower bound we can use minibatches of data, while it is not the case for direct optimization of the acceptance rate. We consider Bayesian inference problem for discriminative model on dataset $\mathcal{D} = \{(x_i, y_i)\}_{i=1}^N$, where $x_i$ is the feature vector of $i$th object and $y_i$ is its label. For the discriminative model we know likelihood $p(y_i \,|\, x_i, \theta)$ and prior distribution $p(\theta)$. In order to obtain predictions for some object $x_i$, we need to evaluate the predictive distribution

$$p(y_i \,|\, x_i) = \mathbb{E}_{p(\theta \,|\, \mathcal{D})}p(y_i \,|\, x_i, \theta). \tag{7}$$

To obtain samples from posterior distribution $p(\theta \,|\, \mathcal{D})$ we suggest to learn proposal distribution $q_\phi(\theta)$ and perform independent MH algorithm. Thus the objective 6 can be rewritten as

$$\mathcal{L}\left( p(\theta \,|\, \mathcal{D}), q_\phi(\theta) \right) = \mathrm{KL}\left( p(\theta' \,|\, \mathcal{D})q_\phi(\theta) \middle\| p(\theta \,|\, \mathcal{D})q_\phi(\theta') \right) \tag{8}$$

Note that due to the usage of independent proposal, the minimized KL-divergence splits up into the sum of two KL-divergences.

$$\mathrm{KL}\left(p(\theta' \,|\, \mathcal{D})q_\phi(\theta) \middle\| p(\theta \,|\, \mathcal{D})q_\phi(\theta')\right) = \mathrm{KL}\left(q_\phi(\theta) \middle\| p(\theta \,|\, \mathcal{D})\right) + \mathrm{KL}\left(p(\theta' \,|\, \mathcal{D}) \middle\| q_\phi(\theta')\right) \quad (9)$$

Minimization of the first KL-divergence corresponds to the variational inference procedure.

$$\mathrm{KL}\left(q_\phi(\theta) \middle\| p(\theta \,|\, \mathcal{D})\right) = -\mathbb{E}_{\theta \sim q_\phi(\theta)} \sum_{i=1}^{N} \log p(y_i \,|\, x_i, \theta) + \mathrm{KL}(q_\phi(\theta)\|p(\theta)) + \log p(\mathcal{D}) \quad (10)$$

The second KL-divergence has the only term that depends on $\phi$. Thus we obtain the following optimization problem

$$\min_\phi \left[ -\mathbb{E}_{\theta \sim q_\phi(\theta)} \sum_{i=1}^{N} \log p(y_i \,|\, x_i, \theta) + \mathrm{KL}(q_\phi(\theta)\|p(\theta)) - \mathbb{E}_{\theta \sim p(\theta \,|\, \mathcal{D})} \log q_\phi(\theta) \right]. \quad (11)$$

The first summand here contains the sum over all objects in dataset $\mathcal{D}$. We follow doubly stochastic variational inference and suggest to perform unbiased estimation of the gradient in problem 11 using only minibatches of data. Moreover, we can use recently proposed techniques (Korattikara et al., 2014; Chen et al., 2016) that perform the independent MH algorithm using only minibatches of data. Combination of these two techniques allows us to use only minibatches of data during iterations of algorithm 1. In the case of the direct optimization of the acceptance rate, straightforward usage of minibatches results in biased gradients. Indeed, for the direct optimization of the acceptance rate (Eq. 1) we have the product over the all training data inside $\min$ function.

## 3.2 SAMPLE-BASED SETTING

In the sample-based setting, we assume the proposal to be an implicit probabilistic model, i.e. the model that we can only sample from. As in the density-based setting, we assume that we are able to perform the reparameterization trick for the proposal.

In this subsection we consider only Markov chain proposal $q_\phi(x' \,|\, x)$, but everything can be applied to independent proposal $q_\phi(x')$ by simple substitution $q_\phi(x' \,|\, x)$ with $q_\phi(x')$. From now we will assume our proposal distribution to be a neural network that takes $x$ as its input and outputs $x'$. Considering proposal distribution parameterized by a neural network allows us to easily exclude delta-function from the space of solutions. We avoid learning the identity mapping by using neural networks with the bottleneck and noisy layers. For the detailed description of the architectures see Appendix C.8.

The set of samples from the true distribution $X \sim p(x)$ allows for the Monte Carlo estimation of the loss

$$\mathcal{L}(p, q_\phi) = \mathbb{E}_{\substack{x \sim p(x) \\ x' \sim q_\phi(x' \,|\, x)}} \log \frac{p(x)q_\phi(x' \,|\, x)}{p(x')q_\phi(x \,|\, x')}. \quad (12)$$

To compute the density ratio $\frac{p(x)q_\phi(x' \,|\, x)}{p(x')q_\phi(x \,|\, x')}$ we suggest to use well-known technique of density ratio estimation via training discriminator network. Denoting discriminator output as $D(x, x')$, we suggest the following optimization problem for the discriminator.

$$\min_D \left[ -\mathbb{E}_{\substack{x \sim p(x) \\ x' \sim q_\phi(x' \,|\, x)}} \log D(x, x') - \mathbb{E}_{\substack{x \sim p(x) \\ x' \sim q_\phi(x' \,|\, x)}} \log(1 - D(x', x)) \right] \quad (13)$$

Speaking informally, such discriminator takes two images as input and tries to figure out which image is sampled from true distribution and which one is generated by the one step of proposal distribution. It is easy to show that optimal discriminator in problem 13 will be

$$D(x, x') = \frac{p(x)q_\phi(x' \,|\, x)}{p(x)q_\phi(x' \,|\, x) + p(x')q_\phi(x \,|\, x')}. \quad (14)$$

Note that for optimal discriminator we have $D(x, x') = 1 - D(x', x)$. In practice, we have no optimal discriminator and these values can differ significantly. Thus, we have four ways for density ratio estimation that may differ significantly.

$$\frac{p(x)q_\phi(x' \,|\, x)}{p(x')q_\phi(x \,|\, x')} \approx \frac{D(x, x')}{1 - D(x, x')} \approx \frac{1 - D(x', x)}{D(x', x)} \approx \frac{1 - D(x', x)}{1 - D(x, x')} \approx \frac{D(x, x')}{D(x', x)} \qquad (15)$$

To avoid the ambiguity we suggest to use the discriminator of a special structure. Let $\widetilde{D}(x, x')$ be a convolutional neural network with scalar output. Then the output of discriminator $D(x, x')$ is defined as follows.

$$D(x, x') = \frac{\exp(\widetilde{D}(x, x'))}{\exp(\widetilde{D}(x, x')) + \exp(\widetilde{D}(x', x))} \qquad (16)$$

In other words, such discriminator can be described as the following procedure. For single neural network $\widetilde{D}(\cdot, \cdot)$ we evaluate two outputs $\widetilde{D}(x, x')$ and $\widetilde{D}(x', x)$. Then we take softmax operation for these values. Summing up all the steps, we obtain algorithm 2.

---

**Algorithm 2** Optimization of proposal distribution in sample-based case

---

**Require:** implicit probabilistic model $q_\phi(x' \,|\, x)$
**Require:** large set of samples $X \sim p(x)$
   **for** $n$ iterations **do**
       sample $\{x_k\}_{k=1}^K \sim X$
       sample $\{x_k'\}_{k=1}^K \sim q_\phi(x'|x)$
       train discriminator $D$ by optimizing 13
       $\mathcal{L}(p, q_\phi) \approx \frac{1}{K} \sum_{k=1}^K \log \frac{D(x_k, x_k')}{1 - D(x_k, x_k')}$        ▷ approximate loss with finite number of samples
       $\phi \leftarrow \phi - \alpha \nabla_\phi \mathcal{L}(p, q_\phi)$        ▷ perform gradient descent step
   **end for**
   **return** parameters $\phi$

---

Algorithm 2 could also be employed for direct optimization of the acceptance rate (Eq. 1). But, in Appendix B.2 we provide an intuition for this setting that the direct optimization of the acceptance rate may struggle from vanishing gradients.

## 4 EXPERIMENTS

In this section, we provide experiments for both density-based and sample-based settings, showing the proposed procedure is applicable to high dimensional target distributions. Code for reproducing all of the experiments will be published with the camera-ready version of the paper.

### 4.1 SYNTHETIC DISTRIBUTIONS

To demonstrate performance of our approach we reproduce the experiment from (Song et al., 2017). For target distributions we use synthetic 2d distributions (see Appendix C.3 for densities): **ring** (a ring-shaped density), **mog2** (a mixture of 2 Gaussians), **mog6** (a mixture of 6 Gaussians), **ring5** (a mixture of 5 distinct rings). We measure performance of learned samplers using *Effective Sample Size* (see Appendix C.4 for formulation). Since the unnormalized densities of target distributions are given, we can learn proposals as suggested in the density-based setting (Section 3.1).

To learn the independent proposal we use RealNVP model (Dinh et al., 2016) (see details in Appendix C.2) and compare the performance of proposals after optimization of different objectives: the acceptance rate (**AR**), our lower bound on the acceptance rate (**ARLB**), evidence lower bound that corresponds to the variational inference (**VI**). We also compare the performance of obtained independent proposals with the performance of Markov chain proposals: **A-NICE-MC** (Song et al., 2017), Hamiltonian Monte Carlo (**HMC**).

In Tables 2, 3 we see that our approach has comparable performance with A-NICE-MC (Song et al., 2017). However, comparison between A-NICE-MC and learning independent proposal is not the main subject of interest, since A-NICE-MC learns Markov chain proposal. On the one hand, Markov

chain proposal uses more information while generating a new sample, hence can learn more expressive stationary distribution, on the other hand, usage of previous sample increase autocorrelation between samples and reduces ESS. Thus, the main point of interest is the comparison of two independent proposals: one is learned by maximization of the acceptance rate (or its lower bound), and the second is learned by variational inference procedure, i.e. maximization of evidence lower bound. In Table 2 we see that both maximization of the acceptance rate and its lower bound outperform variational inference for all target distributions. Moreover, in Fig. 1 we show that variational inference fails to cover all the modes of **mog6** in contrast to proposals learned via maximization of acceptance rate or its lower bound. Densities of learned proposals and histograms for all distributions are presented in Appendix C.6.

Table 2: Performance of learned proposals as measured by Effective Sample Size (see Appendix C.4 for formulation). Higher is better (1000 maximum). All the numbers are rounded to integers. See description of the compared methods in the text.

| Target | A-NICE-MC | HMC | AR (ours) | ARLB (ours) | VI |
|--------|-----------|-----|-----------|-------------|-----|
| ring | **1000** | **1000** | 851 | 850 | 702 |
| mog2 | 355 | 1 | **786** | 604 | 297 |
| mog6 | 320 | 1 | 311 | **367** | 12 |
| ring5 | 156 | 1 | **336** | 249 | 170 |

Table 3: Performance of learned proposals as measured by Effective Sample Size per second. All the numbers are rounded to integers. See description of the compared methods in the text.

| Target | A-NICE-MC | AR (ours) | ARLB (ours) | VI |
|--------|-----------|-----------|-------------|-----|
| ring | **706417** | 275164 | 275488 | 227253 |
| mog2 | 231253 | **267491** | 205552 | 101074 |
| mog6 | **143877** | 94938 | 112034 | 3663 |
| ring5 | 85186 | **95314** | 70635 | 48224 |

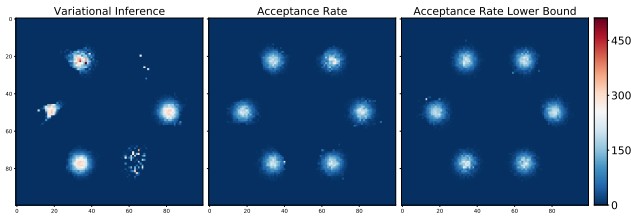

Figure 1: 2d histrogram of samples from the MH algorithm with different proposals. From left to right proposals are learned by: variational inference, the acceptance rate maximization, the acceptance rate lower bound maximization.

## 4.2 DENSITY-BASED SETTING

In density-based setting, we consider Bayesian inference problem for the weights of a neural network. In our experiments we consider approximation of predictive distribution (Eq. 7) as our main goal. To estimate the goodness of the approximation we measure negative log-likelihood and accuracy on the test set.

In subsection 3.1 we show that lower bound on acceptance rate can be optimized more efficiently than acceptance rate due to the usage of minibatches. But other questions arise.

1. Does the proposed objective in Eq. 11 allow for better estimation of predictive distribution compared to the variational inference?

2. Does the application of the MH correction to the learned proposal distribution allow for better estimation of the predictive distribution (Eq. 7) than estimation via raw samples from the proposal?

To answer these questions we consider reduced LeNet-5 architecture (see Appendix C.7) for classification task on 20k images from MNIST dataset (for test data we use all of the MNIST test

set). Even after architecture reduction we still face a challenging task of learning a complex distribution in 8550-dimensional space. For the proposal distribution we use fully-factorized gaussian $q_\phi(\theta) = \prod_{j=1}^{d} \mathcal{N}(\theta_j \mid \mu_j, \sigma_j)$ and standard normal distribution for prior $p(\theta) = \prod_{j=1}^{d} \mathcal{N}(\theta_j \mid 0, 1)$.

For variational inference, we train the model using different initialization and pick the model according to the best ELBO. For our procedure, we do the same and choose the model by the maximum value of the acceptance rate lower bound. In Algorithm 1 we propose to sample from the posterior distribution using the independent MH and the current proposal. It turns out in practice that it is better to use the currently learned proposal $q_\phi(\theta) = \mathcal{N}(\theta \mid \boldsymbol{\mu}, \boldsymbol{\sigma})$ as the initial state for random-walk MH algorithm. That is, we start with the mean $\boldsymbol{\mu}$ as an initial point, and then use random-walk proposal $q(\theta' \mid \theta) = \mathcal{N}(\theta' \mid \theta, \boldsymbol{\sigma})$ with the variances $\boldsymbol{\sigma}$ of current independent proposal. This should be considered as a heuristic that improves the approximation of the loss function.

The optimization of the acceptance rate lower bound results in the better estimation of predictive distribution than the variational inference (see Fig. 2). Optimization of acceptance rate for the same number of epochs results in nearly $30\%$ accuracy on the test set. That is why we do not report results for this procedure in Fig. 2.

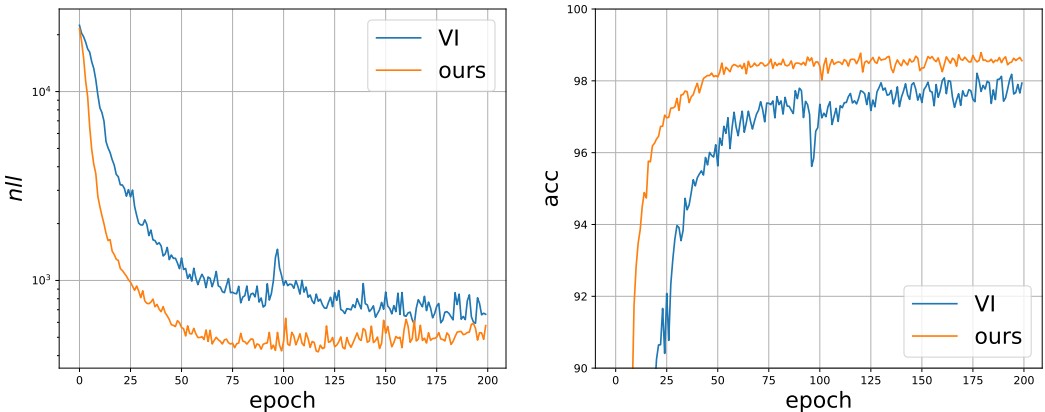

Figure 2: Negative log-likelihood (left) and accuracy (right) on test set of MNIST dataset for variational inference (blue lines) and the optimization of the acceptance rate lower bound (orange lines). In both procedures we apply the independent MH algorithm to estimate the predictive distribution.

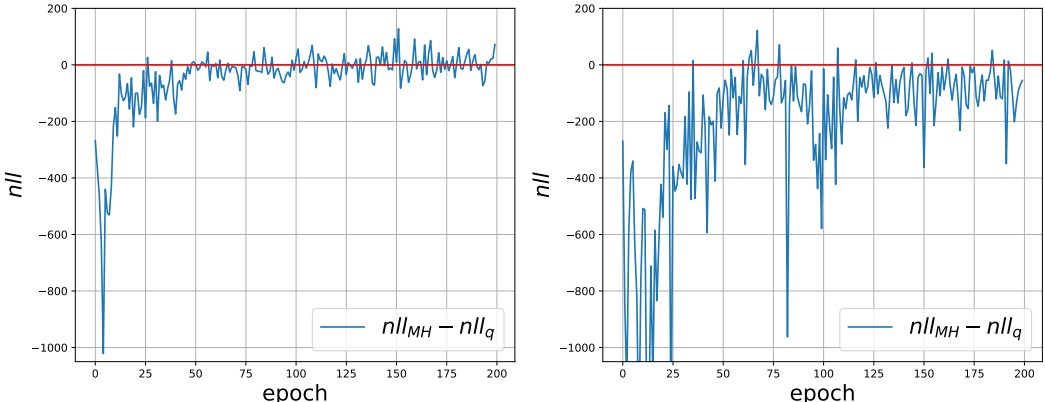

Figure 3: Test negative log-likelihood for two approximations of the predictive distribution based on samples: from proposal distribution $nll_q$ and after MH correction $nll_{MH}$. Left figure corresponds to the optimization of the acceptance rate lower bound, right figure corresponds to the variational inference.

To answer the second question we estimate predictive distribution in two ways. The first way is to perform 100 accept/reject steps of the independent MH algorithm with the learned proposal $q_\phi(\theta)$

after each epoch, i.e. perform MH correction of the samples from the proposal. The second way is to take the same number of samples from $q_\phi(\theta)$ without MH correction. For both estimations of predictive distribution, we evaluate negative log-likelihood on the test set and compare them.

The MH correction of the learned proposal improves the estimation of predictive distribution for the variational inference (right plot of Fig. 3) but does not do so for the optimization of the acceptance rate lower bound (left plot of Fig. 3). This fact may be considered as an implicit evidence that our procedure learns the proposal distribution with higher acceptance rate.

### 4.3 SAMPLE-BASED SETTING

In the sample-based setting, we estimate density ratio using a discriminator. Hence we do not use the minibatching property (see subsection 3.1) of the obtained lower bound, and optimization problems for the acceptance rate and for the lower bound have the same efficiency in terms of using data. That is why our main goal in this setting is to compare the optimization of the acceptance rate and the optimization of the lower bound. Also, in this setting, we have Markov chain proposal that is interesting to compare with the independent proposal. Summing up, we formulate the following questions:

1. Does the optimization of the lower bound has any benefits compared to the direct optimization of the acceptance rate?
2. Do we have mixing issue while learning Markov chain proposal in practice?
3. Could we improve the visual quality of samples by applying the MH correction to the learned proposal?

We use DCGAN architecture for the proposal and discriminator (see Appendix C.8) and apply our algorithm to MNIST dataset. We consider two optimization problems: direct optimization of the acceptance rate and its lower bound. We also consider two ways to obtain samples from the approximation of the target distribution — use raw samples from the learned proposal, or perform the MH algorithm, where we use the learned discriminator for density ratio estimation.

In case of the independent proposal, we show that the MH correction at evaluation step allows to improve visual quality of samples — figures 4(a) and 4(b) for the direct optimization of acceptance rate, figures 4(c) and 4(d) for the optimization of its lower bound. Note that in Algorithm 2 we do not apply the independent MH algorithm during training. Potentially, one can use the MH algorithm considering any generative model as a proposal distribution and learning a discriminator for density ratio estimation. Also, for this proposal, we demonstrate the negligible difference in visual quality of samples obtained by the direct optimization of acceptance rate (see Fig. 4(a)) and by the optimization of the lower bound (see Fig. 4(c)).

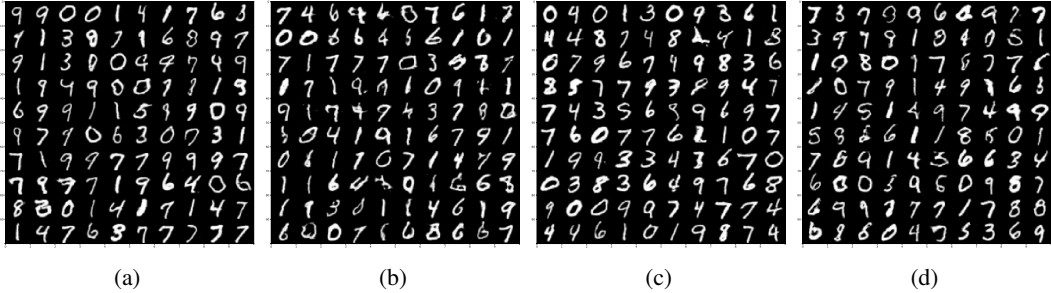

|  (a)  |  (b)  |  (c)  |  (d)  |

Figure 4: Samples from the learned independent proposal obtained via optimization: of acceptance rate (4(a), 4(b)) and its lower bound (4(c), 4(d)). In Fig. 4(b), 4(d) we show raw samples from the learned proposal. In Fig. 4(a), 4(c) we show the samples after applying the independent MH correction to the samples, using the learned discriminator for density ratio estimation.

In the case of the Markov chain proposal, we show that the direct optimization of acceptance rate results in slow mixing (see Fig. 5(a)) — most of the time the proposal generates samples from one of the modes (digits) and rarely switches to another mode. When we perform the optimization of the lower bound the proposal switches between modes frequently (see Fig. 5(b)).

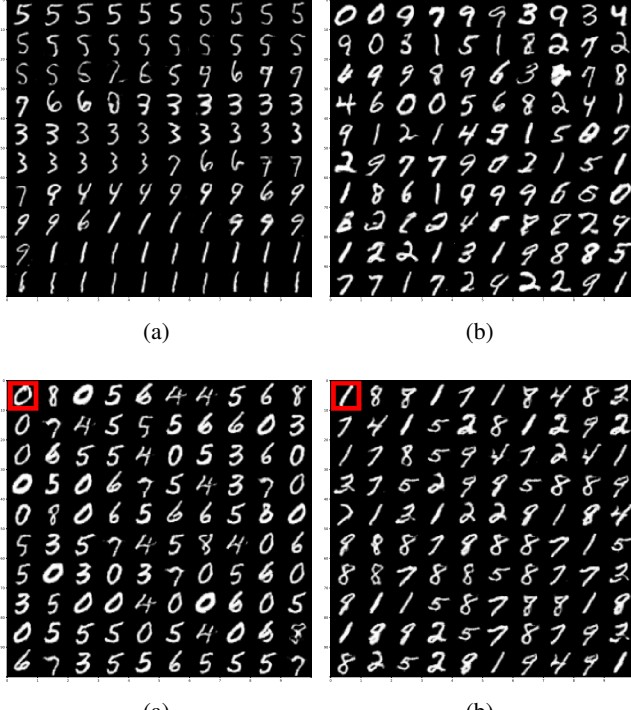

Figure 5: Samples from the chain obtained via the MH algorithm with the learned proposal and the learned discriminator for density ratio estimation. Fig. 5(a) corresponds to the direct optimization of the acceptance rate. Fig. 5(b) – to optimization of the lower bound on acceptance rate. Samples in the chain are obtained one by one from left to right from top to bottom.

(a)     (b)

Figure 6: Samples from the proposal distribution and conditioned on the digit in the red box. The proposal was optimized according to the lower bound on the acceptance rate. Note that we obtain different distributions of the samples because of conditioning of our proposal.

(a)     (b)

To show that the learned proposal distribution has the Markov property rather than being totally independent, we show samples from the proposal conditioned on two different points in the dataset (see Fig. 6). The difference in samples from two these distributions (Fig. 6(a), 6(a)) reflects the dependence on the conditioning.

Additionally, in Appendix C.9 we present samples from the chain after 10000 accepted images and also samples from the chain that was initialized with noise.

## 5    DISCUSSION AND FUTURE WORK

This paper proposes to use the acceptance rate of the MH algorithm as the universal objective for learning to sample from some target distribution. We also propose the lower bound on the acceptance rate that should be preferred over the direct maximization of the acceptance rate in many cases. The proposed approach provides many ways of improvement by the combination with techniques from the recent developments in the field of MCMC, GANs, variational inference. For example

- The proposed loss function can be combined with the loss function from (Levy et al., 2017), thus allowing to learn the Markov chain proposal in the density-based setting.
- We can use stochastic Hamiltonian Monte Carlo (Chen et al., 2014) for the loss estimation in Algorithm 1.
- In sample-based setting one can use more advanced techniques of density ratio estimation.

Application of the MH algorithm to improve the quality of generative models also requires exhaustive further exploration and rigorous treatment.

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

# A ACCEPTANCE RATE OF THE MH ALGORITHM

## A.1 PROOF OF THEOREM 1

Remind that we have random variables $\xi = \frac{p(x')q(x\,|\,x')}{p(x)q(x'\,|\,x)}, x \sim p(x), x' \sim q(x'\,|\,x)$ and $u \sim$ Uniform$[0,1]$, and want to prove the following equalities.

$$\mathbb{E}_\xi \min\{1, \xi\} = \mathbb{P}\{\xi > u\} = 1 - \frac{1}{2}\mathbb{E}_\xi|\xi - 1| \tag{17}$$

Equality $\mathbb{E}_\xi \min\{1, \xi\} = \mathbb{P}\{\xi > u\}$ is obvious.

$$\mathbb{E}_\xi \min\{1, \xi\} = \int_0^\infty p_\xi(x) \min\{1, x\}dx = \int_{x \geq 1} p_\xi(x)dx + \int_{x < 1} p_\xi(x)xdx \tag{18}$$

$$\mathbb{P}\{\xi > u\} = \int_0^\infty dx p_\xi(x) \int_0^x [0 \leq u \leq 1]du = \int_{x \geq 1} p_\xi(x)dx + \int_{x < 1} p_\xi(x)xdx \tag{19}$$

Equality $\mathbb{P}\{\xi > u\} = 1 - \frac{1}{2}\mathbb{E}_\xi|\xi - 1|$ can be proofed as follows.

$$\mathbb{P}\{\xi > u\} = \int_0^1 du \int_u^{+\infty} p_\xi(x)dx = \int_0^1 (1 - F_\xi(u))du = \tag{20}$$

$$= 1 - \left[ uF_\xi(u)\Big|_0^1 - \int_0^1 up_\xi(u)du \right] = 1 - F_\xi(1) + \int_0^1 up_\xi(u)du, \tag{21}$$

where $F_\xi(u)$ is CDF of random variable $\xi$. Note that $F_\xi(0) = 0$ since $\xi \in (0, +\infty]$. Eq. 21 can be rewritten in two ways.

$$1 - F_\xi(1) + \int_0^1 up_\xi(u)du = 1 + \int_0^1 (u - 1)p_\xi(u)du = 1 - \int_0^1 |u - 1|p_\xi(u)du \tag{22}$$

To rewrite Eq. 21 in the second way we note that $\mathbb{E}\xi = 1$.

$$1 - F_\xi(1) + \int_0^1 up_\xi(u)du = \int_1^{+\infty} p_\xi(u)du + 1 - \int_1^{+\infty} up_\xi(u)du = 1 - \int_1^{+\infty} |u - 1|p_\xi(u)du \tag{23}$$

Summing equations 22 and 23 results in the following formula

$$\mathbb{P}\{\xi > u\} = 1 - \frac{1}{2}\mathbb{E}_\xi|\xi - 1|. \tag{24}$$

Using the form of $\xi$ we can rewrite the acceptance rate as

$$1 - \frac{1}{2}\mathbb{E}_\xi|\xi - 1| = 1 - \mathrm{TV}\left( p(x')q(x\,|\,x')\Big\| p(x)q(x'\,|\,x) \right). \tag{25}$$

## A.2 ACCEPTANCE RATE OF INDEPENDENT MH DEFINES SEMIMETRIC IN DISTRIBUTION SPACE

In independent case we have $\xi = \frac{p(x')q(x)}{p(x)q(x')}, x \sim p(x), x' \sim q(x')$ and we want to prove that $\mathbb{E}_\xi|\xi - 1|$ is semimetric (or pseudo-metric) in space of distributions. For this appendix, we denote $D(p, q) = \mathbb{E}_\xi|\xi - 1|$. The first two axioms for metric obviously holds

1. $D(p, q) = 0 \iff p = q$
2. $D(p, q) = D(q, p)$

There is an example when triangle inequality does not hold. For distributions $p = $ Uniform$[0, 2/3], q = $ Uniform$[1/3, 1], s = $ Uniform$[0, 1]$

$$D(p, s) + D(q, s) = \frac{4}{3} < \frac{3}{2} = D(p, q). \tag{26}$$

But weaker inequality can be proved.

$$D(p,s) + D(q,s) = \int |p(x)s(y) - p(y)s(x)|dydx + \int |q(x)s(y) - q(y)s(x)|dydx = \quad (27)$$

$$= \int \left[ |\underbrace{p(x)s(y)q(z)}_{a} - \underbrace{p(y)s(x)q(z)}_{b}| + |\underbrace{q(x)s(y)p(z)}_{c} - \underbrace{q(y)s(x)p(z)}_{d}| \right] dxdydz \quad (28)$$

$$D(p,s) + D(q,s) = \int |p(z)s(y)q(x) - p(y)s(z)q(x)|dxdydz + \quad (29)$$

$$+ \int |q(x)s(z)p(y) - q(z)s(x)p(y)|dxdydz \geq \int \left| \underbrace{q(x)s(y)p(z)}_{c} - \underbrace{p(y)s(x)q(z)}_{b} \right| dxdydz \quad (30)$$

$$D(p,s) + D(q,s) = \int |p(z)s(x)q(y) - p(x)s(z)q(y)|dxdydz + \quad (31)$$

$$+ \int |q(y)s(z)p(x) - q(z)s(y)p(x)|dxdydz \geq \int \left| \underbrace{q(y)s(x)p(z)}_{d} - \underbrace{p(x)s(y)q(z)}_{a} \right| dxdydz \quad (32)$$

Summing up equations 28, 30 and 32 we obtain

$$3(D(p,s) + D(q,s)) \geq \int dxdydz \left[ |a - b| + |c - d| + |c - b| + |d - a| \right] \geq 2 \int dxdydz |d - b| = \quad (33)$$

$$= 2 \int dxdydz s(x) \left| q(y)p(z) - q(z)p(y) \right| = 2D(p,q) \quad (34)$$

$$D(p,s) + D(q,s) \geq \frac{2}{3} D(p,q) \quad (35)$$

# B  OPTIMIZATION OF PROPOSAL DISTRIBUTION

## B.1  ON COLLAPSING TO THE DELTA-FUNCTION

Firstly, let's consider the case of gaussian random-walk proposal $q(x' \mid x) = \mathcal{N}(x' \mid x, \sigma)$. The optimization problem for the acceptance rate takes the form

$$\text{AR} = \int dxdx' p(x)\mathcal{N}(x' \mid x, \sigma) \min\left\{1, \frac{p(x')}{p(x)}\right\} \to \max_{\sigma}. \quad (36)$$

It is easy to see that we can obtain acceptance rate arbitrarly close to 1, taking $\sigma$ small enough.

In the case of the independent proposal, we don't have the collapsing to the delta-function problem. In our work, it is important to show non-collapsing during optimization of the lower bound, but the same hold for the direct optimization of the acceptance rate. To provide such intuition we consider one-dimensional case where we have some target distribution $p(x)$ and independent proposal $q(x) = \mathcal{N}(x \mid \mu, \sigma)$. Choosing $\sigma$ small enough, we approximate sampling with the independent MH as sampling on some finite support $x \in [\mu - a, \mu + a]$. For this support, we approximate the target distribution with the uniform distribution (see Fig. 7).

For such approximation, optimization of lower bound takes the form

$$\min_{q} \left[ \text{KL}(p(x)\|q(x)) + \text{KL}(q(x)\|p(x)) \right] \quad (37)$$

$$\min_{\sigma} \left[ \text{KL}(\text{Uniform}[-a, a]\|\mathcal{N}(x \mid 0, \sigma, -a, a)) + \text{KL}(\mathcal{N}(x \mid 0, \sigma, -a, a)\|\text{Uniform}[-a, a]) \right] \quad (38)$$

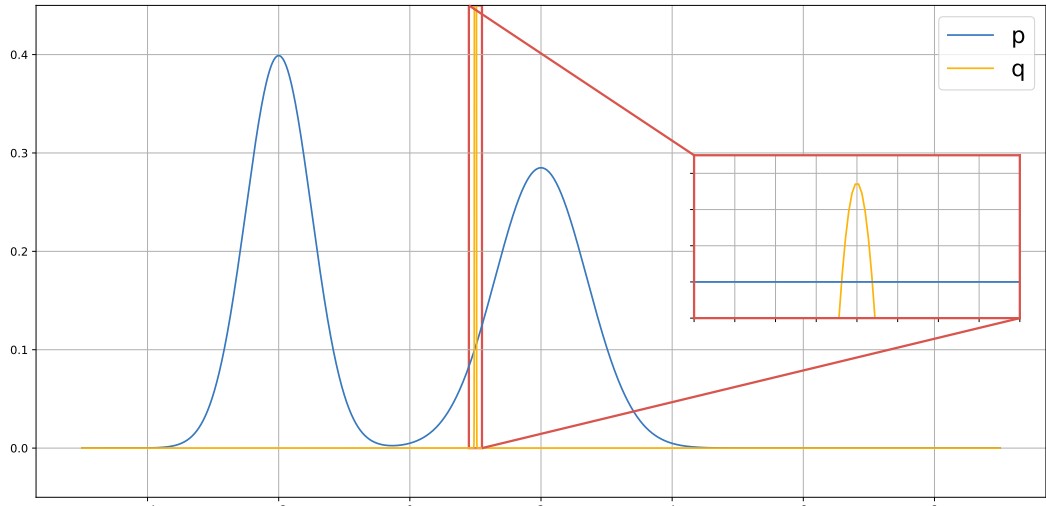

Figure 7: In this figure we show schematic view of approximation of of target distribution with uniform distribution. Red bounding box is made bigger for better comprehension.

Here $\mathcal{N}(x \mid 0, \sigma, -a, a)$ is truncated normal distribution. The first KL-divergence can be written as follows.

$$\mathrm{KL}(\mathrm{Uniform}[-a,a] \| \mathcal{N}(x \mid 0, \sigma, -a, a)) = -\frac{1}{2a} \int_{-a}^{a} dx \log \mathcal{N}(x \mid 0, \sigma, -a, a) - \log 2a = \quad (39)$$

$$= -\frac{1}{2a} \left[ -2a \log(\sigma Z) - a \log 2\pi - \frac{1}{2\sigma^2} \frac{2a^3}{3} \right] - \log 2a = \quad (40)$$

$$= \log \sigma + \log Z + \frac{a^2}{6\sigma^2} + \frac{1}{2} \log 2\pi - \log 2a \quad (41)$$

Here $Z$ is normalization constant of truncated log normal distribution and $Z = \Phi(a/\sigma) - \Phi(-a/\sigma)$, where $\Phi(x)$ is CDF of standard normal distribution. The second KL-divergence is

$$\mathrm{KL}(\mathcal{N}(x \mid 0, \sigma, -a, a) \| \mathrm{Uniform}[-a,a]) = \quad (42)$$

$$= -\frac{1}{2} \log(2\pi e) - \log \sigma - \log Z + \frac{a}{\sqrt{2\pi}\sigma Z} \exp\left(-\frac{a^2}{2\sigma^2}\right) + \log 2a \quad (43)$$

Summing up two KL-divergencies and taking derivative w.r.t. $\sigma$ we obtain

$$\frac{\partial}{\partial \sigma}\left(\mathrm{KL}(\mathrm{Uniform}[-a,a] \| \mathcal{N}(x \mid 0, \sigma, -a, a)) + \mathrm{KL}(\mathcal{N}(x \mid 0, \sigma, -a, a) \| \mathrm{Uniform}[-a,a])\right) = \quad (44)$$

$$= -\frac{a^2}{3\sigma^3} + \frac{a^3}{\sqrt{2\pi}\sigma^4 Z} \exp\left(-\frac{a^2}{2\sigma^2}\right) + \frac{a}{\sqrt{2\pi}} \exp\left(-\frac{a^2}{2\sigma^2}\right)\left[-\frac{1}{\sigma^2 Z} - \frac{1}{\sigma Z^2}\frac{-2a}{\sigma^2\sqrt{2\pi}} \exp\left(-\frac{a^2}{2\sigma^2}\right)\right] = \quad (45)$$

$$= \frac{1}{a}\left[-\frac{a^3}{3\sigma^3} + \frac{a^2}{\sqrt{2\pi}\sigma^2 Z} \exp\left(-\frac{a^2}{2\sigma^2}\right)\left(\frac{a^2}{\sigma^2} - 1 + \frac{2a}{\sqrt{2\pi}\sigma Z} \exp\left(-\frac{a^2}{2\sigma^2}\right)\right)\right] \quad (46)$$

To show that the derivative of the lower bound w.r.t. $\sigma$ is negative, we need to prove that the following inequality holds for positive $x$.

$$-\frac{1}{3}x^3 + \frac{x^2}{\sqrt{2\pi}(\Phi(x) - \Phi(-x))} \exp(-x^2/2)\left(x^2 - 1 + \frac{2x}{\sqrt{2\pi}(\Phi(x) - \Phi(-x))} \exp(-x^2/2)\right) < 0, \quad x > 0 \quad (47)$$

Defining $\phi(x) = \int_0^x e^{-t^2/2}dt$ and noting that $2\phi(x) = \sqrt{2\pi}(\Phi(x) - \Phi(-x))$ we can rewrite inequality 47 as

$$\frac{1}{\phi(x)} e^{-x^2/2}\left(x^2 - 1 + \frac{2xe^{-x^2/2}}{\phi(x)}\right) < \frac{2x}{3}, \quad x > 0 \quad (48)$$

By the fundamental theorem of calculus, we have

$$xe^{-x^2/2} = \int_0^x e^{-t^2/2}(1-t^2)dt \tag{49}$$

Hence,

$$\phi(x) - xe^{-x^2/2} = \int_0^x e^{-t^2/2}t^2 dt \geq e^{-x^2/2}\int_0^x t^2 dt = e^{-x^2/2}\frac{x^3}{3} \tag{50}$$

Or equivalently,

$$\phi(x) \geq e^{-x^2/2}\frac{x^3 + 3x}{3} \tag{51}$$

Using this inequality twice, we obtain

$$\frac{e^{-x^2/2}}{\phi(x)} \leq \frac{3}{x(x^2+3)} \tag{52}$$

and

$$x^2 - 1 + \frac{xe^{-x^2/2}}{\phi(x)} \leq x^2 - 1 + \frac{3}{x^2+3} = \frac{x^2(2+x^2)}{x^2+3} \tag{53}$$

Thus, the target inequality can be verified by the verification of

$$\frac{3x(2+x^2)}{(x^2+3)^2} \leq \frac{2x}{3}. \tag{54}$$

Thus, we show that partial derivative of our lower bound w.r.t. $\sigma$ is negative. Using that knowledge we can improve our loss by taking a bigger value of $\sigma$. Hence, such proposal does not collapse to delta-function.

## B.2 Intuition for better gradients in sample-based setting

In this section, we provide an intuition for sample-based setting that the loss function for lower bound has better gradients than the loss function for acceptance rate. Firstly, we remind that in the sample-based setting we use a discriminator for density ratio estimation.

$$D(x, x') = \frac{p(x)q(x'\,|\,x)}{p(x)q(x'\,|\,x) + p(x')q(x\,|\,x')} \tag{55}$$

For this purpose we use the discriminator of special structure

$$D(x, x') = \frac{\exp(\widetilde{D}(x, x'))}{\exp(\widetilde{D}(x, x')) + \exp(\widetilde{D}(x', x))} = \frac{1}{1 + \exp\left(-(\widetilde{D}(x, x') - \widetilde{D}(x', x))\right)} \tag{56}$$

We denote $d(x, x') = \widetilde{D}(x, x') - \widetilde{D}(x', x)$ and consider the case when the discriminator can easily distinguish fake pairs from valid pairs. So $D(x, x')$ is close to 1 and $d(x, x') \gg 0$ for $x \sim p(x)$ and $x' \sim q(x'\,|\,x)$. To evaluate gradients we consider Monte Carlo estimations of each loss and take gradients w.r.t. $x'$ in order to obtain gradients for parameters of proposal distribution. We do not introduce the reparameterization trick to simplify the notation but assume it to be performed. For the optimization of the acceptance rate we have

$$\int dxdx' p(x)q(x'\,|\,x)\left|\frac{p(x')q(x\,|\,x')}{p(x)q(x'\,|\,x)} - 1\right| \simeq \left|\frac{p(x')q(x\,|\,x')}{p(x)q(x'\,|\,x)} - 1\right| \tag{57}$$

$$L_{\text{AR}} = \left|\frac{p(x')q(x\,|\,x')}{p(x)q(x'\,|\,x)} - 1\right| \approx \left|\frac{1 - D(x, x')}{D(x, x')} - 1\right| \tag{58}$$

$$\frac{\partial L_{\text{AR}}}{\partial x'} = \frac{1}{D^2(x, x')}\frac{\partial D(x, x')}{\partial x'} = \exp(-d(x, x'))\frac{\partial d(x, x')}{\partial x'} \tag{59}$$

While for the optimization of the lower bound we have

$$\int dx dx' p(x) q(x' \,|\, x) \log \left( \frac{p(x) q(x' \,|\, x)}{p(x') q(x \,|\, x')} \right) \simeq \log \left( \frac{p(x) q(x' \,|\, x)}{p(x') q(x \,|\, x')} \right) \tag{60}$$

$$L_{\text{LB}} = -\log \left( \frac{p(x') q(x \,|\, x')}{p(x) q(x' \,|\, x)} \right) \approx -\log \left( \frac{1 - D(x, x')}{D(x, x')} \right) \tag{61}$$

$$\frac{\partial L_{\text{LB}}}{\partial x'} = \frac{1}{(1 - D(x, x')) D(x, x')} \frac{\partial D(x, x')}{\partial x'} = \frac{\partial d(x, x')}{\partial x'} \tag{62}$$

Now we compare Eq. 59 and Eq. 62. We see that in case of strong discriminator we have vanishing gradients in Eq. 59 due to $\exp(-d(x, x'))$, while it is not the case for Eq. 62.

## C   EXPERIMENTS

### C.1   TOY PROBLEM

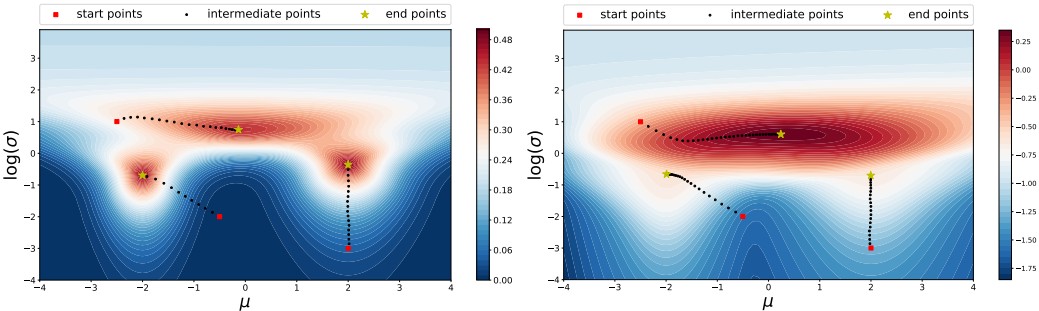

Figure 8: Level-plots in parameter space for the toy problem. Left: level-plot for the acceptance rate of the MH algorithm. Right: level-plot for the lower bound of the acceptance rate.

This experiment shows that it is possible to optimize the acceptance rate, optimizing its lower bound. For the target distribution we consider bimodal Gaussian $p(x) = 0.5 \cdot \mathcal{N}(x \,|\, -2, 0.5) + 0.5 \cdot \mathcal{N}(x \,|\, 2, 0.7)$, for the independent proposal we consider unimodal gaussian $q(x) = \mathcal{N}(x \,|\, \mu, \sigma)$. We perform stochastic gradient optimization using Algorithm 1 from the same initialization for both objectives (Fig. 8) and obtain approximately the same local maximums.

### C.2   ARCHITECTURE OF THE REALNVP PROPOSAL

For the proposal distribution we use similar architecture to the NICE proposal. The RealNVP model (Dinh et al., 2016) use the same strategy for evaluating the Jacobian as the NICE model does. Each *coupling layer* define the following function. Given a $D$ dimensional input $x$ and $d < D$, the output $y$ is evaluated by the formula

$$y_{1:d} = x_{1:d},$$
$$y_{d+1:D} = x_{d+1:D} \odot \exp(s(x_{1:d})) + t(x_{1:d}),$$

where the functions $s, t$ can be arbitrary complex, since the structure of the functions doesn't influence the computation of the Jacobian.

For our proposal we use $4$ coupling layers with $s$ and $t$ consist of two fully-connected layers with hidden dimension of $256$.

### C.3   SYNTHETIC DISTRIBUTIONS DENSITIES

For synthetic distributions we consider the same distributions as in Song et al. (2017).

The analytic form of $p(x)$ for *ring* is:

$$p(x) \propto \exp(-U(x)), \quad U(x) = \frac{(\sqrt{x_1^2 + x_2^2} - 2)^2}{0.32} \tag{63}$$

The analytic form of $p(x)$ for *mog2* is:

$$p(x) = \frac{1}{2}\mathcal{N}(x|\mu_1, \sigma_1) + \frac{1}{2}\mathcal{N}(x|\mu_2, \sigma_2) \tag{64}$$

where $\mu_1 = [5, 0]$, $\mu_2 = [-5, 0]$, $\sigma_1 = \sigma_2 = [0.5, 0.5]$.

The analytic form of $p(x)$ for *mog6* is:

$$p(x) = \frac{1}{6}\sum_{i=1}^{6}\mathcal{N}(x|\mu_i, \sigma_i) \tag{65}$$

where $\mu_i = [\sin\frac{i\pi}{3}, \cos\frac{i\pi}{3}]$ and $\sigma_i = [0.5, 0.5]$.

The analytic form of $p(x)$ for *ring5* is:

$$p(x) \propto \exp(-U(x)), \quad U(x) = \min(u_1, u_2, u_3, u_4, u_5) \tag{66}$$

where $u_i = (\sqrt{x_1^2 + x_2^2} - i)^2/0.04$.

### C.4 EFFECTIVE SAMPLE SIZE FORMULATION

For the effective sample size formulation we follow Song et al. (2017).

Assume a target distribution $p(x)$, and a Markov chain Monte Carlo (MCMC) sampler that produces a set of N correlated samples $\{x_i\}_1^N$ from some distribution $q(\{x_i\}_1^N)$ such that $q(x_i) = p(x_i)$. Suppose we are estimating the mean of $p(x)$ through sampling; we assume that increasing the number of samples will reduce the variance of that estimate.

Let $V = \text{Var}_q[\sum_{i=1}^{N} x_i/N]$ be the variance of the mean estimate through the MCMC samples. The effective sample size (ESS) of $\{x_i\}_1^N$, which we denote as $M = ESS(\{x_i\}_1^N)$, is the number of independent samples from $p(x)$ needed in order to achieve the same variance, i.e. $\text{Var}_p[\sum_{j=1}^{M} x_j/M] = V$. A practical algorithm to compute the ESS given $\{x_i\}_1^N$ is provided by:

$$ESS(\{x_i\}_1^N) = \frac{N}{1 + 2\sum_{s=1}^{N-1}(1 - \frac{s}{N})\rho_s} \tag{67}$$

where $\rho_s$ denotes the autocorrelation under $q$ of $x$ at lag $s$. We compute the following empirical estimate $\hat{\rho}_s$ for $\rho_s$:

$$\hat{\rho}_s = \frac{1}{\hat{\sigma}^2(N-s)}\sum_{n=s+1}^{N}(x_n - \hat{\mu})(x_{n-s} - \hat{\mu}) \tag{68}$$

where $\hat{\mu}$ and $\hat{\sigma}$ are the empirical mean and variance obtained by an independent sampler.

Due to the noise in large lags $s$, we adopt the approach of Hoffman & Gelman (2014) where we truncate the sum over the autocorrelations when the autocorrelation goes below 0.05.

### C.5 OPTIMIZATION OF THE LOWER BOUND

In this section we provide the empirical evidence that maximization of the proposed lower bound on the acceptance rate (**ARLB**) results in maximization of the acceptance rate (**AR**). For that purpose we evaluate **ARLB** and **AR** at each iteration during the optimization of **ARLB**. After training we evaluate correlation coefficient between **ARLB** and logarithm of **AR**. The curves are shown in Fig. 9. Correlation coefficients for different distributions are: $-0.914$ (**ring**), $-0.905$ (**mog2**), $-0.956$ (**mog6**), $-0.982$ (**ring5**).

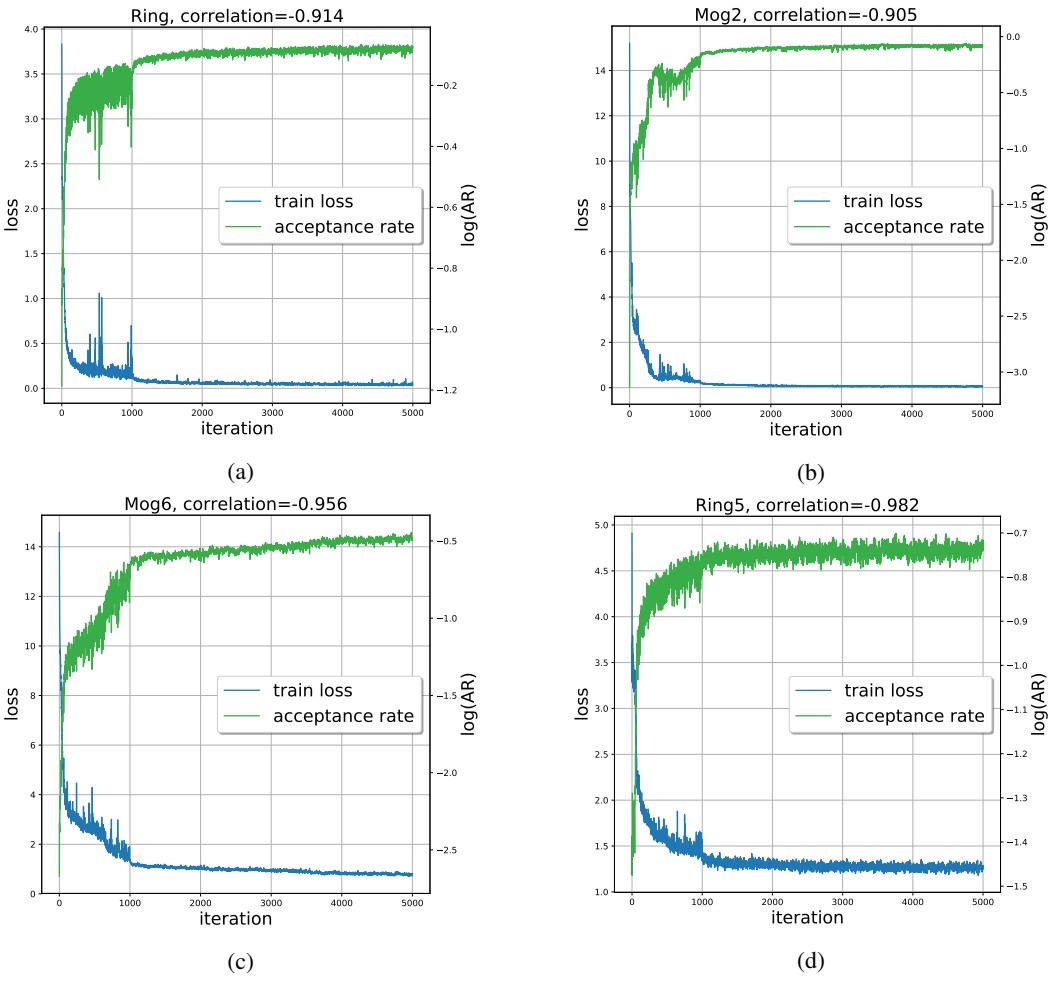

Figure 9: plots for the acceptance rate and the acceptance rate lower bound evaluated at every iteration during the optimization of the acceptance rate lower bound. Correlation coefficient is evaluated between the logarithm of the acceptance rate and the acceptance rate lower bound.

## C.6 LEARNED PROPOSALS

In this section we provide levelplots of learned proposals densities (see Fig. 10). We also provide 2d histrograms of samples from the MH algorithm using the corresponding proposals (see Fig. 11).

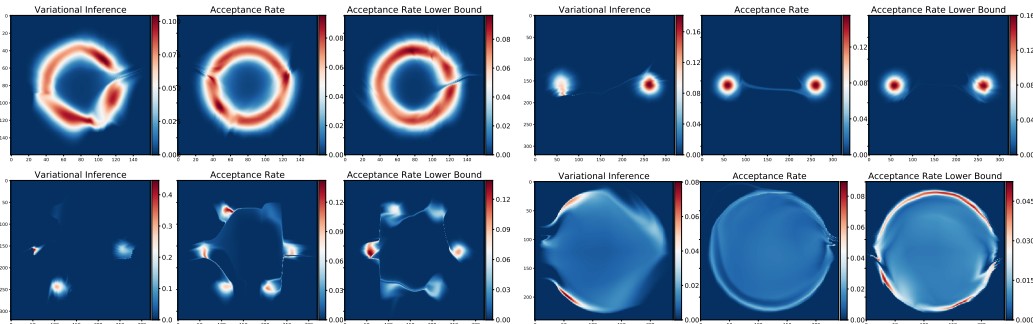

Figure 10: levelplots of learned proposal densities. For each distribution from left to right proposals are learned by: variational inference, the acceptance rate maximization, the acceptance rate lower bound maximization.

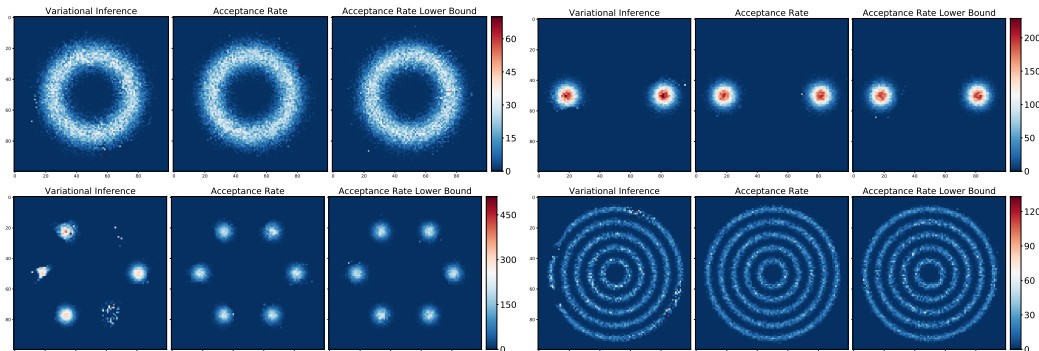

Figure 11: 2d histrograms of samples from the MH algorithm with different proposals. For each distribution from left to right proposals are learned by: variational inference, the acceptance rate maximization, the acceptance rate lower bound maximization.

## C.7 ARCHITECTURE OF THE REDUCED LENET-5

```
class LeNet5(BayesNet):
    def __init__(self):
        super(LeNet5, self).__init__()
        self.num_classes = 10
        self.conv1 = layers.ConvFFG(1, 10, 5, padding=0)
        self.relu1 = nn.ReLU(True)
        self.pool1 = nn.MaxPool2d(2, padding=0)
        self.conv2 = layers.ConvFFG(10, 20, 5, padding=0)
        self.relu2 = nn.ReLU(True)
        self.pool2 = nn.MaxPool2d(2, padding=0)
        self.flatten = layers.ViewLayer([20*4*4])
        self.dense1 = layers.LinearFFG(20*4*4, 10)
        self.relu3 = nn.ReLU()
        self.dense2 = layers.LinearFFG(10, 10)
```

### C.8 ARCHITECTURES OF NEURAL NETWORKS IN SAMPLE-BASED SETTING

In sample-based setting we use usual DCGAN architecture for independent proposal distribution

```
class Generator(layers.ModuleWrapper):
    def __init__(self):
        super(Generator, self).__init__()
        self.fc = nn.Linear(100, 128*8*8)
        self.unflatten = layers.ViewLayer([128, 8, 8])
        self.in1 = nn.InstanceNorm2d(128)
        self.us1 = nn.ConvTranspose2d(128, 128, 2, 2)
        self.conv1 = nn.Conv2d(128, 128, 3, stride=1, padding=1)
        self.in2 = nn.InstanceNorm2d(128, 0.8)
        self.lrelu1 = nn.LeakyReLU(0.2, inplace=True)
        self.us2 = nn.ConvTranspose2d(128, 128, 2, 2)
        self.conv2 = nn.Conv2d(128, 64, 3, stride=1, padding=1)
        self.in3 = nn.InstanceNorm2d(64, 0.8)
        self.lrelu2 = nn.LeakyReLU(0.2, inplace=True)
        self.conv3 = nn.Conv2d(64, 1, 3, stride=1, padding=1)
        self.tanh = nn.Tanh()
```

And a little be modified acrhitecture for Markov chain proposal distribution

```
class Generator(layers.ModuleWrapper):
    def __init__(self):
        super(Generator, self).__init__()

        self.d_conv1 = nn.Conv2d(1, 16, 5, stride=2, padding=2)
        self.d_lrelu1 = nn.LeakyReLU(0.2, inplace=True)
        self.d_do1 = nn.Dropout2d(0.5)
        self.d_conv2 = nn.Conv2d(16, 4, 5, stride=2, padding=2)
        self.d_in2 = nn.InstanceNorm2d(4, 0.8)
        self.d_lrelu2 = nn.LeakyReLU(0.2, inplace=True)
        self.d_do2 = nn.Dropout2d(0.5)

        self.b_view = layers.ViewLayer([4*8*8])
        self.b_fc = nn.Linear(4*8*8, 256)
        self.b_lrelu = nn.LeakyReLU(0.2, inplace=True)
        self.b_fc = nn.Linear(256, 128 * 8 * 8)
        self.b_do = layers.AdditiveNoise(0.5)

        self.e_unflatten = layers.ViewLayer([128, 8, 8])
        self.e_in1 = nn.InstanceNorm2d(128, 0.8)
        self.e_us1 = nn.ConvTranspose2d(128, 128, 2, 2)
        self.e_conv1 = nn.Conv2d(128, 128, 3, stride=1, padding=1)
        self.e_in2 = nn.InstanceNorm2d(128, 0.8)
        self.e_lrelu1 = nn.LeakyReLU(0.2, inplace=True)
        self.e_us2 = nn.ConvTranspose2d(128, 128, 2, 2)
        self.e_conv2 = nn.Conv2d(128, 64, 3, stride=1, padding=1)
        self.e_in3 = nn.InstanceNorm2d(64, 0.8)
        self.e_lrelu2 = nn.LeakyReLU(0.2, inplace=True)
        self.e_conv3 = nn.Conv2d(64, 1, 3, stride=1, padding=1)
        self.e_tanh = nn.Tanh()
```

For both proposals we use the proposed discriminator with the following architecture.

```
class Discriminator(nn.Module):
    def __init__(self):
        super(Discriminator, self).__init__()
        self.conv1 = nn.Conv2d(2, 16, 3, 2, 1)
```

```
        self.lrelu1 = nn.LeakyReLU(0.2, inplace=True)
        self.conv2 = nn.Conv2d(16, 32, 3, 2, 1)
        self.lrelu2 = nn.LeakyReLU(0.2, inplace=True)
        self.in2 = nn.InstanceNorm2d(32, 0.8)
        self.conv3 = nn.Conv2d(32, 64, 3, 2, 1)
        self.lrelu3 = nn.LeakyReLU(0.2, inplace=True)
        self.in3 = nn.InstanceNorm2d(64, 0.8)
        self.conv4 = nn.Conv2d(64, 128, 3, 2, 1)
        self.lrelu4 = nn.LeakyReLU(0.2, inplace=True)
        self.in4 = nn.InstanceNorm2d(128, 0.8)
        self.flatten = layers.ViewLayer([128*2*2])
        self.fc = nn.Linear(128*2*2, 1)

    def forward(self, x, y):
        xy = torch.cat([x, y], dim=1)
        for module in self.children():
            xy = module(xy)
        yx = torch.cat([y, x], dim=1)
        for module in self.children():
            yx = module(yx)
        return F.softmax(torch.cat([xy, yx], dim=1), dim=1)
```

## C.9 Additional figures for Markov chain proposals in sample-based setting

In this section, we show additional figures for Markov chain proposals. In Fig. 12 we show samples from the chain that was initialized by the noise. In Fig. 13 we show samples from the chain after 10000 accepted samples.

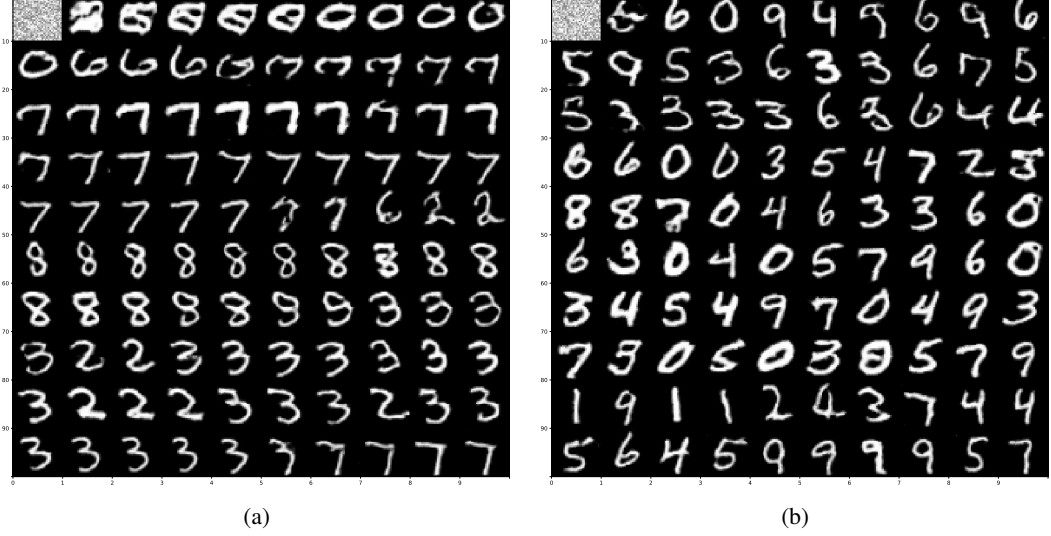

(a)  (b)

Figure 12: Samples from the chain initialized with noise. To obtain samples we use the MH algorithm with the learned proposal and the learned discriminator for density ratio estimation. In Fig. 5(a) we use proposal and discriminator that are learned during optimization of acceptance rate. In Fig. 5(b) we use proposal and discriminator that are learned during the optimization of the acceptance rate lower bound. Samples in the chain are obtained one by one from left to right from top to bottom starting with noise (first image in the figure).

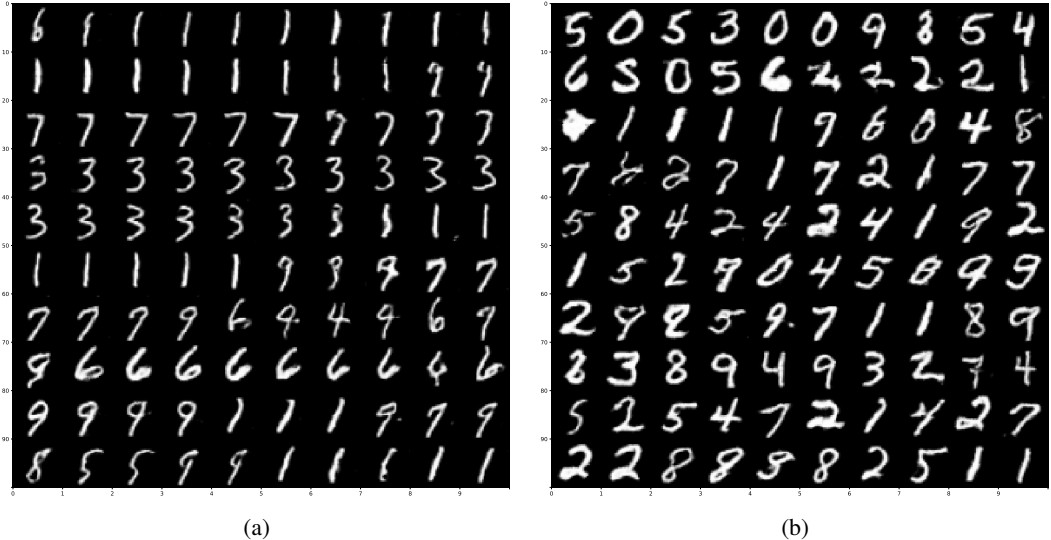

(a)                                         (b)

Figure 13: Samples from the chain after 10000 accepted samples. To obtain samples we use the MH algorithm with the learned proposal and the learned discriminator for density ratio estimation. In Fig. 5(a) we use proposal and discriminator that are learned during optimization of acceptance rate. In Fig. 5(b) we use proposal and discriminator that are learned during the optimization of the acceptance rate lower bound. Samples in chain are obtained one by one from left to right from top to bottom.

