# OpenReview forum: "Metropolis-Hastings view on variational inference and adversarial training"
_ICLR.cc/2019/Conference_

### Official Review · AnonReviewer2 · 2018-11-02
**Some points must clarified**

**Rating:** 9
**Confidence:** 4

**Review:**

I think the paper could be published, however I have some concerns.

Mayor comments:

- My main concern is that I do not understand why do not directly apply the  KL divergence with respect to p(x) and q(x) instead of considering p(x)\times q(x') and  p(x')\times q(x). More specifically, I have understood that your approach is motivated by Theorem 1 (nice result, by the way) but I am not sure it is better than just applying the KL divergence with respect to p(x) and q(x), directly.

- The state-of-the-art discussion for MCMC schemes in the introduction must be completed at least including the Multiple Try Metropolis algorithms,

J. S. Liu, F. Liang, W. H. Wong, The multiple-try method and local optimization in metropolis sampling, Journal of the American Statistical Association 95 (449) (2000) 121–134.

L. Martino, "A Review of Multiple Try MCMC algorithms for Signal Processing", Digital Signal Processing, Volume 75, Pages: 134-152, 2018.

The sentence about adaptive MCMC's should be also completed.

Minor comments:

- Why do you say that "MCMC is non-parametric" in the introduction? in which sense? MCMC methods are sampling algorithms. Please, clarify.

- In my opinion, Eq. (5)-(6)-(8)-(9)-(11)-(12)-(13) are not proper mathematically written  (maybe the same "wrong" way of written that, is repeated  in other parts of the text).

- The results in the Figures in the simulations should be averaged more. Specially, Figure 3.

---

> ### Author Response · Authors · 2018-11-26
> **Thanks for the review!**
>
>
> Thank you for your valuable feedback!
>
> > My main concern is that I do not understand why do not directly apply the KL divergence with respect to p(x) and q(x) instead of considering p(x)\times q(x') and p(x')\times q(x).
>
> As you pointed out, this objective is motivated by maximization of the acceptance rate, since our final goal is to obtain efficient proposal for the Metropolis-Hastings algorithm.
> We added new experiments in the paper (Section 4.1) where we compared the performance of proposals learned using different objectives. We additionally performed the comparison of proposed objective with KL(p||q) on synthetic data in terms of Effective Sample Size:
> Ring:                                       820 for KL(p||q), 850 for KL(q||p) + KL(p||q)
> Mixture of 2 gaussians:      466 for KL(p||q), 604 for KL(q||p) + KL(p||q)
> Mixture of 6 gaussians:      336 for KL(p||q), 367 for KL(q||p) + KL(p||q)
> Mixture of 5 distinct rings: 291 for KL(p||q), 255 for KL(q||p) + KL(p||q)
> Since the sampling from proposal distribution is much easier than sampling from the target distribution additional optimization of KL(q||p) does not seem to be very expensive. Moreover, the Monte Carlo estimation of KL(q||p) has much less variance during the first iterations of the algorithm and allows to improve the initial proposal faster.
>
> > The state-of-the-art discussion for MCMC schemes in the introduction must be completed...
>
> Thanks for these papers. We have cited them in the new version of our paper.
>
> > Why do you say that "MCMC is non-parametric" in the introduction? in which sense?
>
> You are right, this formulation is confusing. We have reformulated this sentence in the new version.
>
> > In my opinion, Eq. (5)-(6)-(8)-(9)-(11)-(12)-(13) are not proper mathematically written.
>
> We have improved the notation in the new version.

---

### Official Review · AnonReviewer3 · 2018-11-02
**The theoretic finding that maximizing the Metropolis Hastings acceptance rate is equivalent to minimize the symmetric KL divergence between target and proposal is great. Though, I have doubts about the correctness of the (main) proposed algorithm leveraging those insights and would expect, at least, an empirical verification of correctness.**

**Rating:** 6
**Confidence:** 4

**Review:**

One of the main contributions of the paper is showing how maximizing the acceptance rate in Metropolis Hastings (MH) translates in minimizing the symmetric KL divergence of the target and the proposal distribution. The result aligns with the intuition that if this variance is 0, detailed balanced is satisfied and, hence, we can always accept the proposal. Also Equation 11 nicely fits the intuition that a good (independent) proposal should minimize the KL divergence between proposal and target (as in VI) under the constraint that the proposal has full support compared to the target distribution, which is enforced by the last term. Theorem 1 and its proof are great.

However, the proposed algorithms leveraging these findings are problematic. Algorithm 1 suggest independent Metropolis-Hastings in order to avoid the proposal to collapse to a point distribution, that is, a Dirac delta function centered at the current position. However, in the experiments, the authors study a "modified" version using a random walk proposal parameterized with the current (diagonal) covariance estimate. This is surprising as the authors explicitly avoided this proposal when motivating MH acceptance rate maximization.

In any case, the correctness of the algorithm is neither shown for an independent proposal nor a Markov Chain proposal. Indeed, I would argue that Algorithm 1 generally does not find a sample distributed according to the target distribution. The algorithm assumes that we can create a sample from the target p(x) that can be used to approximate the loss (a bound on the expected acceptance rate). However, if we could find an unbiased sample in the very first iteration of the algorithm, we could immediately stop and there wouldn't be a need for the algorithm at all. Hence, we must assume that the sample drawn from the target is biased (in the beginning); which is indeed a realistic assumption as neither independent MH nor random walk MH will yield an unbiased sample in any reasonable time (for any practical problem). However, this would bias the loss and, consequently, the parameter update for the proposal distribution. In particular, for random walk MH, I would expect the covariance to contract such that the final proposal indeed collapses to a point distribution. This is because the proposal is only required to have support around the observed samples and this area will become smaller over time. I would expect a proof that the sample at iteration k is "better" than a sample drawn at iteration k-1, to show that the bias vanishes over time. Though, I assume that this is hard to show as the proposal parameters form a Markov Chain itself. So at least a rigor empirical study is needed.

Therefore, I would expect a metric measuring the quality of the final sample. The log-likelihood is not such a measure. While the marginalized log-likelihood could measure the quality of the sample, we cannot compute it for any real data/model (which is why we use sampling in the first place). So we need some artificial settings. However, the 1-dimensional toy example is insufficient as MH primarily suffers in high-dimensional spaces. It would be interesting to also report the acceptance rate depending on the number of dimensions of the target distribution. I would assume an exponential decay; even with learning, which might be the reason why the authors only report random walk MH in Section 4.2.

Algorithm 2 does not require to sample from some target distribution but can leverage the observed sample. While the algorithm nicely connects GANs and sampling, the actual value of the algorithm is not fully clear to me. Learning an independent proposal reduces the problem to learning a GAN; and learning a Markov Chain seems only relevant for sampling-based inference; however, we already have a sample from the target distribution, and we can sample more data using a trained GAN.

Minor comments:
- The prefix/in-fix notation of integrals is mixed, e.g. in Eq 19, "dx" appears before the integrand, but "du" appears after the integrand of the inner integral.


UPDATE:

The revised version is much better in empirically demonstrating the value of the method; though, there is still some work needed. First, the artificial examples are still rather low-dimensional where independent MH is expected to perform well. Second, ESS does not help to assess the biasedness of the sample; maybe [1] can help with this. Third, NUTS might be a better baseline than standard HMC which is know to be sensitive to the stepsize/number of leapfrog steps. An effective sample size of 1 suggests that the method did not even start to sample - likely because of a bad choice of the stepsize and/or mass matrix. I would suggest using PyMC3's NUTS implementation. Finally, to show the correctness of the method, I would suggest to 1) let alpha converge to zero such that \phi will be fixed at some point, and 2) ensure that the proposal has full support under the target for any value of \phi. In this case, the sample drawn from the target will be unbiased for large enough n (same arguments as for adaptive MCMC should apply).

The idea of reusing the samples from previous iterations for approximating the loss is interesting and worth exploring.

[1] Jackson Gorham, Lester Mackey. "Measuring Sample Quality with Kernels", https://arxiv.org/abs/1703.01717

---

> ### Author Response · Authors · 2018-11-26
> **Thanks for the review!**
>
>
> Thank you for providing such detailed feedback!
>
> > However, the proposed algorithms leveraging these findings are problematic...
>
> We use random walk proposal only to obtain the samples from the target distribution during the training. Thus, we don’t learn the Markov chain proposal, but use it to estimate the loss for the independent proposal. Test performance is reported for samples obtained from the independent proposal. Using the random walk proposal can be considered as heuristic that allows to obtain better samples from the posterior during the training.
>
> > In any case, the correctness of the algorithm is neither shown for an independent proposal nor a Markov Chain proposal...
>
> First of all, we’ve provided new experiments (see Section 4.1) that show how the proposed algorithm performs for synthetic distributions. We also demonstrate learned proposals and samples from them (see Appendix C.6) for distributions with distant modes. Our results show that proposals learned by Algorithm 1 cover all the modes of the target distribution while variational inference fails to cover them.
>
> The problem you’ve described can be partially addressed by the choice of the initial variance of the proposal. Making the proposal wide enough allows us to cover all the target distribution and obtain diverse set of samples (most likely, with low values of densities). However, we can reuse these samples on the following iterations, thus preventing our proposal from collapsing to the single mode. Moreover, in Section 4.2 we do not aim to accurately cover full posterior with the single Gaussian (which seems to be infeasible). Our intuition was that in such situation KL(p||q)-term allows us to spread the proposal on several modes, thus describing the posterior better than variational inference does.
>
> > Therefore, I would expect a metric measuring the quality of the final sample...
>
> We provide such artificial setting in the new experiments (Section 4.1). We simulate the situation of distant modes by taking the mixture of six distant Gaussians and show that using the RealNVP model [1] as the independent proposal allows to efficiently sample from this distribution. Using the RealNVP model as the proposal allows to deal with high dimensional distributions - experiments in [1] show that RealNVP can efficiently sample images from ImageNet that has 64x64 resolution. Our approach allows to efficiently train such models in case when the unnormalized density of the target distribution is given.
>
> > Algorithm 2 does not require to sample from some target distribution but can leverage the observed sample...
>
> Similar to GANs, the main purpose of Algorithm 2 is to learn sampler. Besides the connection of GANs and optimization of the acceptance rate of the MH algorithm, Algorithm 2 proposes the extensions to the GANs framework. The minor extension is the Markov chain proposal and formulation of the loss function for it. The major extension to the GANs framework is the usage of the discriminator while sampling to approximately perform the MH correction. In Fig. 4 we demonstrate that such approximate correction allows to obtain samples of better quality.
>
> 1. Laurent Dinh, Jascha Sohl-Dickstein, and Samy Bengio.  Density estimation using real nvp. arXiv preprint arXiv:1605.08803, 2016.

---

> ### Author Response · Authors · 2018-11-29
> **Thanks for the update!**
>
>
> Thank you again for your efforts on reviewing our paper!
>
> > First, the artificial examples are still rather low-dimensional where independent MH is expected to perform well.
>
> Indeed, scaling for a large number of dimensions is important to verify. In order to do that, one needs to learn a sophisticated proposal. Since it seems infeasible to sample from the true posterior distribution using a gaussian as proposal, if we somehow measured the quality of samples obtained via a simple proposal it would be poor. For now, we can say that our model behaves reasonably well in high-dimensions as it allows to obtain better test log-likelihood than variational inference.
>
> > Second, ESS does not help to assess the biasedness of the sample; maybe [1] can help with this.
> > Finally, to show the correctness of the method, I would suggest to 1) let alpha converge to zero such that \phi will be fixed at some point, and 2) ensure that the proposal has full support under the target for any value of \phi. In this case, the sample drawn from the target will be unbiased for large enough n (same arguments as for adaptive MCMC should apply).
>
> Thanks for this paper, we will definitely check it out.
> However, it’s worth mentioning that in the updated version we have added the following paragraph in Section 3.1 to motivate the usage of normalizing flows.
> “In this paper, we consider two types of explicit proposals: simple parametric family (Section 4.2) and normalizing flows (Rezende & Mohamed, 2015; Dinh et al., 2016) (Section 4.1). Rich family of normalizing flows allows to learn expressive proposal and evaluate its density in any point of target distribution space. Moreover, an invertible model (such as normalizing flow) is a natural choice for the independent proposal due to its ergodicity. Indeed, choosing the arbitrary point in the target distribution space, we can obtain the corresponding point in the latent space using the inverse function. Since every point in the latent space has positive density, then every point in the target space also has positive density.”
> Hence, the usage of an ergodic proposal combined with the Metropolis-Hastings algorithm guarantees unbiased sampling from the target distribution after convergence to the stationary distribution. However, the number of steps to obtain a representative set of samples from the target distribution can differ significantly, and ESS allows to estimate such number. Moreover, in Fig. 1 we provide histograms for different proposals and show that qualitative difference of histograms is successfully captured by the quantitative difference in ESS (see Table 1).
>
> > Third, NUTS might be a better baseline than standard HMC which is know to be sensitive to the stepsize/number of leapfrog steps. An effective sample size of 1 suggests that the method did not even start to sample - likely because of a bad choice of the stepsize and/or mass matrix. I would suggest using PyMC3's NUTS implementation.
>
> Our main competitor in terms of ESS is A-NICE-MC method [1], which also learns the neural network as proposal. In this paper they designed such synthetic distributions that are hard for HMC to sample from. For such synthetic distributions HMC fails to jump across modes because they are too distant with low variance, hence gradients around the one of modes have little information about the other modes. These distributions and performance of HMC are reported in [1], maybe it’s better to remove the comparison with HMC from Table 1 so as not to confuse the reader.
>
> 1. Jiaming Song, Shengjia Zhao, and Stefano Ermon. A-nice-mc: Adversarial training for mcmc. In Advances in Neural Information Processing Systems, pp. 5140–5150, 2017.

---

### Official Review · AnonReviewer1 · 2018-11-05
**Empirical evaluation lacking to test the validity of the proposed objective**

**Rating:** 5
**Confidence:** 3

**Review:**

The paper proposes to learn transition kernels for MCMC by optimizing the acceptance rate (or its lower bound) of Metropolis-Hastings.

My main reason for worry is the use of independent proposals for this particular learning objective. While I can buy the argument in the Appendix on how this avoids the collapse problem of Markov proposals, I think the rate of convergence of the chain would greatly reduce in practice because of this assumption.

Unfortunately, the empirical evaluation in this work is lacking to formally confirm or reject my hypothesis. In particular, it is absolutely crucial to compare the performance of this method with Song et al., 2017 (which does not make this assumption) using standard metrics such as Effective Sample Size. Another recent work [1] optimizes for the expected square jump distance and should also have been compared against.

[1]: Levy, Daniel, Matthew D Hoffman, and Jascha Sohl-Dickstein. 2017. “Generalizing Hamiltonian Monte Carlo with Neural Networks.” ArXiv Preprint ArXiv:1711.09268.

---

> ### Author Response · Authors · 2018-11-26
> **Thanks for the review!**
>
>
> Thank you for pointing out this flaw.
> To make things more clear, we added new experiments (see Section 4.1), where we demonstrate how to adapt the expressive family of  RealNVP [1] to learn independent proposals. In these experiments we demonstrate comparable performance with A-NICE-MC [2] model and outperform the proposal learned by variational inference.
> Unfortunately, we have not found the exact parameters of synthetic distributions of L2HMC [3] to compare against it. However, the reduction of autocorrelation is the inherent property of both models. L2HMC reduces autocorrelation by the maximization of the expected square jump distance, while our approach reduces autocorrelation by maximization of the acceptance rate of an independent proposal. Moreover, the combination of two losses (the acceptance rate and expected square jump distance) seems to be the great point of interest for the future research, since the maximization of jump distance won’t allow Markov chain proposal to collapse to the delta-function.
>
> 1. Laurent Dinh, Jascha Sohl-Dickstein, and Samy Bengio.  Density estimation using real nvp. arXiv preprint arXiv:1605.08803, 2016.
> 2. Jiaming Song, Shengjia Zhao, and Stefano Ermon.  A-nice-mc:  Adversarial training for mcmc.  In Advances in Neural Information Processing Systems, pp. 5140–5150, 2017.
> 3. Levy, Daniel, Matthew D Hoffman, and Jascha Sohl-Dickstein. 2017. “Generalizing Hamiltonian Monte Carlo with Neural Networks.” ArXiv Preprint ArXiv:1711.09268.

---

### Author Response · Authors · 2018-11-26
**We have added the requested evaluation**


As reviewers requested, we’ve added additional experiments (Section 4.1).
The new experiments demonstrate performance of our approach on synthetic data for independent proposals. For independent proposal we’ve chosen RealNVP[1] model and have shown that it can be easily adapted for maximization of the acceptance rate. We demonstrate that for all target distributions obtained samplers outperform the proposal with the same architecture but learned via variational inference. We also demonstrate that obtained samplers obtain the comparable performance in terms of Effective Sample Size (ESS) and ESS per second.

Moreover, in Appendix C.5 we provide additional empirical evidence that maximization of the proposed lower bound results in the maximization of the acceptance rate. For that purpose we evaluate the acceptance rate lower bound (ARLB) and the acceptance rate (AR) at each iteration during the optimization of (ARLB). Then we estimate how the ARLB correlates with the logarithm of AR during training. Correlation coefficients for different distributions are:
1. 0.914 for ring distribution,
2. 0.905 for mixture of two gaussians,
3. 0.956 for mixture of six gaussians,
4. 0.982 for mixture of 5 distinct rings.

1. Laurent Dinh, Jascha Sohl-Dickstein, and Samy Bengio.  Density estimation using real nvp. arXiv preprint arXiv:1605.08803, 2016.

---

### Public Comment · ~Anirudh_Goyal1 · 2018-11-28
**You can (also) directly parameterize the transition operator of MCMC chain.**

Hello,

I enjoyed reading your paper.

I would like to point our paper where we learn a transition operator of MCMC chain by directly parameterizing it with a neural network. (Variational Walkback, https://arxiv.org/abs/1711.02282). We feel like our paper should be cited.

If you indeed decide to cite my paper, you should also reference this paper.

[1] https://arxiv.org/abs/1503.03585, Deep Unsupervised Learning using Non equilibrium thermodynamics.

Thanks for your time! :)

---

> ### Public Comment · (anonymous) · 2018-12-01
> **Does variational walk back consider Metropolis Hastings?**
>
> If not, then I think it is not very relevant, since the Markov chain does not have a discrete accept step in VW.

---

> ### Author Response · Authors · 2018-12-02
> **Thanks for your interest!**
>
>
> We are very pleased with your interest in our paper!
> We also were enjoyed by reading yours! :)
>
> It is very important for us to provide the reader with a comprehensive overview of different sampling methods, so we will glad to cite both papers in the camera-ready version of our paper.

---

### Public Comment · (anonymous) · 2018-12-01
**Some questions**

I find the density based formulation interesting and potentially useful for training amortized approximations to energy based models. However, I have some concerns about the experiments and the sample based method.

MH is notoriously slow in large scale datasets because it requires computing the likelihood over the entire dataset. In the MNIST classification experiments, how is Metropolis-Hastings performed? Is this the reason why the dataset is delibrately limited to 20,000 examples? How does this method compare with other stochastic gradient MCMC methods, such as SGLD?

https://arxiv.org/abs/1505.05424 , a VI method, claims significantly better test accuracy than what is claimed here, but it uses a larger dataset, so the results are not directly comparable. Why not use the 50,000-sample MNIST dataset in this paper?

Moreover, the proposed method for sample based has appeared in Song's A-NICE-MC paper (section 3). In fact, it is a special case of Song's method, where m = 1 and x ~ p(x) and converging from random noise is not concerned.

---

> ### Author Response · Authors · 2018-12-02
> **Thanks for your interest!**
>
>
> Thank you for reading our paper!
> Let us make things more clear.
>
> > MH is notoriously slow in large scale datasets because it requires computing the likelihood over the entire dataset. In the MNIST classification experiments, how is Metropolis-Hastings performed? Is this the reason why the dataset is delibrately limited to 20,000 examples? How does this method compare with other stochastic gradient MCMC methods, such as SGLD?
>
> In section 3.1 we show that recently proposed techniques [1,2] can be adapted to use only minibatches of data while sampling via the Metropolis-Hastings algorithm. It allows us to estimate gradients of the proposed lower bound on the acceptance rate using only minibatches of data. We use only 20 000 of train images in order to make the MNIST classification problem more challenging since it is known that a large number of algorithms are able to obtain almost perfect test classification using the full trainset. For this problem, we compare our approach only with the variational inference as the most popular way of deriving the approximation for posterior that is further used for estimation of predictive distribution.
>
> > https://arxiv.org/abs/1505.05424 , a VI method, claims significantly better test accuracy than what is claimed here, but it uses a larger dataset, so the results are not directly comparable. Why not use the 50,000-sample MNIST dataset in this paper?
>
> In our paper, we use a reduced LeNet-5 architecture with 8550 parameters. In the paper [3] authors use the network from the paper [4] that consists of two fully-connected layers (4000 and 2000 hidden units) with 11136000 parameters in total. We use only 20 000 of train images in order to make the MNIST classification problem more challenging since it is known that a large number of algorithms are able to obtain almost perfect test classification using the full trainset.
>
> > Moreover, the proposed method for sample based has appeared in Song's A-NICE-MC paper (section 3). In fact, it is a special case of Song's method, where m = 1 and x ~ p(x) and converging from random noise is not concerned.
>
> Our algorithm proposed in the sample-based setting is motivated by the maximization of the acceptance rate and entirely derived from the desired objective. The proposed algorithm differs from A-NICE-MC [5] in some key points:
> - We use different loss for the discriminator and the different structure of the discriminator. The discriminator in A-NICE-MC tries to distinguish between (real, real) pairs and (fake, real) pairs, while our discriminator tries to distinguish between (fake, real) pairs and (real, fake) pairs. That implies different loss function and results in the estimation of density ratio that is needed for the acceptance rate estimation.
> - We use different loss for the generator (both in case of maximization of the acceptance rate and its lower bound). The generator loss in A-NICE-MC is motivated by [6]. In our paper, we propose to learn the generator to maximize the acceptance rate (or its lower bound).
> - We propose to use the discriminator while sampling for the approximate density ratio estimation and performing the MH algorithm.
> Hence, our algorithm can’t be considered as a special case of A-NICE-MC.
>
> 1. Korattikara, Anoop, Yutian Chen, and Max Welling. "Austerity in MCMC land: Cutting the Metropolis-Hastings budget." In International Conference on Machine Learning, pp. 181-189. 2014.
> 2. Chen, Haoyu, Daniel Seita, Xinlei Pan, and John Canny. "An Efficient Minibatch Acceptance Test for Metropolis-Hastings." arXiv preprint arXiv:1610.06848 (2016).
> 3. Blundell, Charles, Julien Cornebise, Koray Kavukcuoglu, and Daan Wierstra. "Weight uncertainty in neural networks." arXiv preprint arXiv:1505.05424 (2015).
> 4. Nair, Vinod, and Geoffrey E. Hinton. "Rectified linear units improve restricted boltzmann machines." In Proceedings of the 27th international conference on machine learning (ICML-10), pp. 807-814. 2010.
> 5. Jiaming Song, Shengjia Zhao, and Stefano Ermon. A-nice-mc: Adversarial training for mcmc. In Advances in Neural Information Processing Systems, pp. 5140–5150, 2017.
> 6. Arjovsky, Martin, Soumith Chintala, and Léon Bottou. "Wasserstein gan." arXiv preprint arXiv:1701.07875 (2017).

---

### Meta-Review · Area_Chair1 · 2018-12-11
**Nice theoretic finding and connection of GAN with MCMC, but some concerns are to be addressed for a strong publication**

**Confidence:** 4
**Recommendation:** Reject

**Metareview:**

This paper provides a good finding that maximizing a lower bound of the M-H acceptance rate is equivalent to minimizing the symmetric KL divergence between target the proposal. This lower bound is then used to learn sampler for both density and sample-based settings. It also nicely connects GAN with MCMC by providing a novel loss function to train the discriminator. Experiment on MNIST dataset in Sec 4.2 shows training the proposal with the symmetric KL is better than variational inference that optimizes KL(q||p).

However, there are a few concerns raised in both the reviews and other comments that should be further clarified.
1. Training an independent proposal may reduce the rate of convergence.
2. In the density-based setting experiments, the learnt independent proposal is only used to provide an initial point and a random-walk kernel is actually used for sampling. This is different from what is proposed algorithm in Section 3.
3. The proposed algorithm is only compared with VI in density-based setting, and there are no comparison with other baselines in the sample-based setting, despite the close connections of the proposed method with other models. Stochastic gradient MCMC methods, A-NICE-MC, GAN will be good baselines for empirical comparisons. Also, the dataset in Sec 4.2 is a subset of the standard MNIST, which makes comparison with other literatures difficult.

For the first concern, the authors provided new experiments for low-dimensional synthetic distributions. It is very helpful to show the comparable performance with A-NICE-MC in this case, but the real challenge in high-dimensional distributions remains unexamined. For the second concern, the authors consider the use of random-walk as a heuristic that allows to obtain better samples from the posterior, but that significantly changes the proposed transition kernel in Alg. 1.

This paper would be significantly stronger and make a very good contribution to this area by addressing the problems above.